# Better Quality Pretraining Data and T5 Models for African Languages

**Akintunde Oladipo[1], Mofetoluwa Adeyemi[1], Orevaoghene Ahia[2], Odunayo Ogundepo[1],**
**Abraham Toluwalase Owodunni[3], David Ifeoluwa Adelani[3,4], Jimmy Lin[1]**

[1] University of Waterloo     [2] University of Washington
[3] Masakhane     [4] University College London

aooladip@uwaterloo.ca

## Abstract

In this study, we highlight the importance of enhancing the quality of pretraining data in multilingual language models. Existing web crawls have demonstrated quality issues, particularly in the context of low-resource languages. Consequently, we introduce a new multilingual pretraining corpus for 16 African languages, designed by carefully auditing existing pretraining corpora to understand and rectify prevalent quality issues. To compile this dataset, we undertake a rigorous examination of current data sources for thirteen languages within one of the most extensive multilingual web crawls, mC4, and extract cleaner data through meticulous auditing and improved web crawling strategies. Subsequently, we pretrain a new T5-based model on this dataset and evaluate its performance on multiple downstream tasks. Our model demonstrates better downstream effectiveness over existing pretrained models across four NLP tasks, underscoring the critical role data quality plays in pretraining language models in low-resource scenarios. Specifically, on cross-lingual QA evaluation, our new model is more than twice as effective as multilingual T5. All code, data and model are publicly available at https://github.com/castorini/AfriTeVa-keji.

## 1 Introduction

As language models have scaled up in size and multilingual capability in recent years, commensurate effort has followed to curate pretraining data (Raffel et al., 2020) to support this growth and improve the alignment of language models.

Earlier multilingual models such as mBERT (Devlin et al., 2019) and XLM-R (Conneau et al., 2019) were trained on monolingual data from Wikipedia and/or other large-scale web crawls which included only a few African languages. The introduction of mC4 (Xue et al., 2021), a document-level dataset spanning 101 languages helped alleviate this cover-

age gap.[1] However, previous work (Kreutzer et al., 2022) has shown that mC4 and other existing large-scale pretraining corpora have numerous quality issues, particularly for the low-resource African languages they contain.

Against this backdrop, indigenous efforts to build language resources for African languages have converged to two approaches: (1) **Small high-quality data** (e.g., 1GB) pretraining where most data are from the clean or verified sources like news domain (Ogueji et al., 2021). (2) **Large aggregation of all available data** (e.g., $15 - 42$ GB) from noisy or unverified sources like CC-100 (Conneau et al., 2020), and mC4, combined with high-quality sources like news corpora (Adelani et al., 2022; Alabi et al., 2022; Adebara et al., 2022).

This tradeoff between quantity and quality is forced by the unavailability of large, quality pretraining data for African languages. Motivated by this need, we introduce a new multilingual pretraining corpus in 20 African languages. We draw from Kreutzer et al. (2022)'s audit of existing pretraining corpora to understand prevailing quality issues. For mC4, they cite a high ratio both of sentences in incorrect languages (15.98% average) and nonlinguistic content (11.40% average). We trace these issues to the quality of data sources used in mC4 for the languages in our study and design heuristics to effectively extract clean monolingual text.

More notably, we demonstrate how large-scale web crawls and document-level datasets, such as mC4, can be enhanced through meticulous auditing of their document sources i.e., base URLs (e.g., www.voahausa.com). Interestingly, for numerous credible sources, mC4 encompasses fewer documents than what is actually available. We conduct our own web crawl of these sources, collecting more documents than what is present in mC4 for

---

[1] While OSCAR (Suarez et al., 2019; Abadji et al., 2022) includes 6 African languages, three of them have roughly 1000 documents. All 6 languages amount to less than 200MB

the respective languages. We consolidate the result of our efforts (cleaning and crawling) with data from other sources, notably Wikipedia, and include four high-resource languages – Arabic, English, French & Portuguese.

To evaluate the quality of our new corpus, we pretrain a new T5-based LM on the collected dataset and benchmark its performance on multiple downstream tasks. Our model demonstrates improved effectiveness over existing pretrained LMs further highlighting the importance of carefully curated datasets for pretraining language models in low-resource scenarios. Our model was significantly better than the baseline mT5 models across four different downstream tasks. Specifically, on cross-lingual QA evaluation, our new model achieves more than double the performance of multilingual T5.

## 2 WURA Dataset

We present WURA,[2] a multilingual dataset comprising 16 African languages and 4 high-resource languages popularly spoken on the African continent – Arabic, English, French, and Portuguese.

The curation of WURA was carried out in a three-part process: – (i) Auditing and cleaning mC4 (ii) Crawling indigenous websites and (iii) Combination with existing language resources.

### 2.1 Auditing and Cleaning mC4

#### 2.1.1 Language Contamination

Kreutzer et al. (2022) reports mC4's high ratio of non-linguistic content and sentences in incorrect languages, with African languages being of particular concern. The authors report significant loss (up to 50%) in recall of correct in-language sentences as they increased precision of their automatic language classification.

Our manual audit of mC4 corroborates the documented issues. We highlight three important findings: (1) The distribution of mC4 document sources has a long tail. Many individual news publications yield thousands of documents in the mC4. (2) Documents from news publications are more likely to be of higher quality i.e., both in-language and grammatical compared to documents from other web sources. (3) Some documents are from websites which translate content using online translation tools. Such documents are often a mix

of in-language and noisy or non-linguistic text, and may best be filtered at sentence-level. Noting all of these issues and findings, we filter at three levels:

**Corpus-level.** We first rank unique websites in descending order of the number of documents they contribute to the mC4 corpus for each language. Then, we select the top 20% of websites for each language and collect documents sourced from websites in this list. This preserves high potential sources for further document level filtering.

**Document-level.** At document level, we filter out documents that do not contain at least 5 stop-words in them (Caswell et al., 2020) using stop-words from Stopword Lists for African Languages dataset.[3]

**Passage-level.** After document-level filtering, we chunk the dataset into passages of roughly 512 tokens. Finally, we filter out passages that contain fewer than 4 unique words or contain repetition for more than 20% of its word length; have more than 40% of its characters are numeric or contain markers of possibly offensive content such as included in the Toxicity-200 dataset (NLLB Team et al., 2022) for the relevant language.

While Kreutzer et al. (2022)'s audit of mC4 did not yield a significant amount of offensive content (0.06% of sentences they audited) and our web crawls mainly focused on verified news publications, these filters ensure that non-linguistic and offensive contents are removed at the passage level.

#### 2.1.2 mC4 is a Great Source!

Xue et al. (2021)'s inclusion of the URL each document is sourced from makes the mC4 corpus even more useful as a data source. Commonly, multiple articles are collected from the same base website, e.g., news publications. For many news publications that provide a sitemap, we find that there are fewer articles in mC4 than is actually available on the websites. Further, mC4 only covers up to August, 2020 so updating the crawls up to the current day yields more data.

We initiate focused crawls for such websites and this leads to significant increase (> 100% for Hausa and Somali) in the amount of articles available per language. For all languages we consider except Chichewa, Sesotho, Xhosa and Zulu, we collect 1.39M articles (see Table 6) from credible sources found in mC4.

---

[2]Wura means Gold in Yoruba – with more refining, the quality of our data and model improves.

[3]https://www.kaggle.com/datasets/rtatman/stopword-lists-for-african-languages

| Model | Size | amh | eng | fra | hau | ibo | lin | lug | orm | pcm | run | sna | som | swa | tir | xho | yor | AVG | AVG$^{SL}$ |
|---|---|---|---|---|---|---|---|---|---|---|---|---|---|---|---|---|---|---|---|
| AfriTeVa-base | 229M | 87.0 | 80.3 | 71.9 | 85.8 | 79.9 | 82.8 | 60.2 | 82.9 | 95.2 | 80.0 | 84.4 | 58.0 | 80.7 | 55.2 | 69.4 | 86.4 | 77.5 | 78.4 |
| mT5-base | 580M | 78.2 | 89.8 | 59.0 | 82.7 | 76.8 | 80.8 | 75.0 | 79.2 | 96.1 | 85.7 | 90.4 | 75.0 | 76.1 | 65.1 | 71.8 | 86.2 | 79.2 | 78.6 |
| FlanT5-base | 580M | 54.5 | **92.4** | **88.9** | 84.5 | **86.6** | **90.6** | 84.1 | 85.8 | **97.8** | 87.3 | 90.6 | 76.0 | 79.0 | 41.5 | 90.8 | 88.9 | 82.5 | 83.2 |
| AfriMT5-base | 580M | 90.2 | 90.3 | 87.4 | 87.9 | 88.0 | 88.6 | 84.8 | 83.9 | 96.6 | 91.0 | 91.5 | **77.8** | 84.4 | 80.8 | 91.6 | 88.8 | 87.7 | 87.8 |
| AfriTeVa V2 | 428M | **92.8** | 90.6 | 88.0 | **89.4** | 86.1 | 86.0 | **91.1** | **90.8** | 96.8 | **92.3** | **93.3** | 75.7 | **87.0** | **86.4** | **93.6** | **92.3** | **89.5** | **88.9** |

Table 1: MasakhaNews classification results: Evaluation is done using the weighted F1 score and the scores presented are averaged across 3 seeds. AfriTeVa V2 surpasses mT5-base by up to 10 points. The average scores excluding languages not in the mC4 corpus are also provided in AVG$^{SL}$.

## 2.2 Combination with Existing Language Resources and Non-African Languages

Following previous works (Alabi et al., 2022; Adebara et al., 2022), we include certain non-African languages in our pretraining data. Specifically, we include over 240,000 articles newly crawled from 10 African news websites reporting in English, French and Portuguese. We also include a sample of 1.5M Wikipedia articles for English and French, as well as Wikipedia articles written in Egyptian Arabic. For the African languages, we include all Wikipedia articles. Finally, we deduplicate using the document URLs. In doing this, we prioritize news articles in our focused crawls over their existing counterparts in mC4.

**Final Dataset Statistics** Table 6 presents a statistical summary of our dataset. The combined dataset from crawling, combining with existing sources and deduplication amounts to ∼30GB of data across all languages and ∼19GB for African languages.

## 3 Experimental Setup

### 3.1 Model

Using t5x and seqio (Roberts et al., 2022), we pretrain a T5 (Shazeer, 2020; Raffel et al., 2020) model with a subword-tokenizer of vocabulary size 150,000. We pretrain for 524,288 steps on the span-corruption objective using the Adafactor optimizer. Each training batch consists of 512 examples, each with an input of 512 tokens and an output of 114 tokens. Our new model is known as AfriTeVa V2, a 428M parameter model.

### 3.2 Downstream Tasks

#### 3.2.1 Cross-lingual Question Answering

We evaluated our models on the test set of AfriQA Ogundepo et al. (2023), a cross-lingual question answering dataset with questions in 10 African languages and gold passages in English or French. We evaluated in zero-shot generative

cross-lingual QA settings using in-lang queries and the provided gold passages in English.

#### 3.2.2 Machine Translation

We evaluated using MAFAND-MT (Adelani et al., 2022) − a machine translation benchmark in the news domain. MAFAND-MT contains few thousand parallel training sentences (2,500-30,000 sentences) for 16 African languages, ideal for evaluating the effective adaptation of pretrained LMs to new languages and domains.

#### 3.2.3 Summarization

For summarization, we use XL-Sum (Hasan et al., 2021), an abstractive summarization dataset which covers 44 languages, including 9 African languages. The authors establish strong baselines on both low and high-resource languages in the dataset through multilingual finetuning of mT5.

#### 3.2.4 Text Classification

We use the news topic classification dataset recently introduced by Adelani et al. (2023) for 16 African languages, MasakhaNews. The authors establish multiple baselines on the dataset using both classical machine learning models and finetuning or prompting language models.

### 3.3 Baseline Models

We compare our new model, AfriTeVa V2, with the base variants of existing multilingual T5 models: mT5 (Xue et al., 2021), ByT5 (Xue et al., 2022) and FlanT5 (Chung et al., 2022), as well as Africentric models: AfriTeVa (Ogundepo et al., 2022), AfriMT5 & AfriByT5 (Adelani et al., 2022).

mT5 was pretrained on the mC4 corpus which is the starter point for this work while ByT5 is the byte-level adaptation of the mT5 model. FlanT5 is T5 instruction-finetuned for improved performance. AfriTeVa, AfriMT5 and AfriByT5 models provide a closer comparison given the nature and focus of our research. While AfriTeVa is a T5 model pretrained on a small corpus (∼1GB), AfriMT5 &

| | en-xx | | | | | | | xx-en | | | | | | |
|---|---|---|---|---|---|---|---|---|---|---|---|---|---|---|
| Model | hau | ibo | pcm | swa | yor | zul | AVG | hau | ibo | pcm | swa | yor | zul | AVG |
| mT5-base | 2.8 | 18.0 | **34.1** | 25.1 | 4.8 | 11.7 | 16.1 | 5.8 | 18.9 | **42.2** | 29.5 | 12.3 | 22.4 | 21.9 |
| AfriMT5-base | 5.1 | 19.6 | 35.0 | 26.7 | 6.2 | 13.2 | 17.5 | 10.4 | 19.5 | 44.6 | 30.6 | 13.8 | 24.0 | 23.8 |
| ByT5-base | 8.3 | 21.8 | 30.1 | 24.4 | 7.5 | 14.0 | 17.7 | 12.9 | 21.0 | 39.4 | 27.1 | 11.5 | 22.8 | 22.5 |
| AfriByT5-base | 9.3 | **22.7** | 30.0 | 24.7 | 7.6 | 15.3 | 18.3 | 13.5 | **20.7** | 39.5 | 27.0 | 11.9 | 24.0 | 22.8 |
| AfriTeVa V2 | **13.4** | 20.7 | 31.1 | **28.0** | **12.1** | **15.6** | **20.3** | **16.2** | 16.7 | 40.5 | **31.0** | **17.6** | **28.4** | **25.1** |

Table 2: MAFAND-MT results: Evaluation is done using the BLEU score and we obtain significantly better performance on average across all languages in both the *en-xx* and *xx-en* directions, except for ibo and pcm.

AfriByT5 are adapted from mT5 and ByT5 models using continual pretraining. Apart from AfriTeVa, AfriTeVa V2 has ∼26% less parameters than the other baseline models.

## 4 Result and Discussion

### 4.1 Downstream Performance

In this section, we compare AfriTeVa V2 to baseline models on selected tasks. For each downstream task, we evaluate under the same conditions. We performed per-language finetuning for machine translation & text classification, multilingual finetuning over 35K steps for summarization.

#### 4.1.1 Cross-lingual Question Answering:

AfriTeVa V2 achieves very impressive results in the cross-lingual question-answering task, especially for languages in our pretraining data. We finetune on the train set of Squad 2.0 (Rajpurkar et al., 2016) dataset and evaluate the models performance on the test set AfriQA. We compare performance on generative gold passage answer prediction, with in-language queries and English passages. Table 4 shows that AfriTeVa V2 achieves much better F1 scores and Exact Match accuracies (∼2×) across 6 out of 7 languages compared to using mT5-Base as the back-bone model.

#### 4.1.2 Machine Translation

We observe higher BLEU scores when translating from African languages into English than in the reverse direction. According to Table 2, we achieve a better score on average, topping mT5 and AfriMT5 base models by ∼1-3 points. While both ByT5-style models show greater effectiveness over the mT5 models, AfriTeVa V2 consistently improves over both results for all languages except ibo and pcm, an English-based creole language.

#### 4.1.3 Summarization

We perform multilingual training for 35,000 steps and sample each batch from a single language. Table 3 shows we match the performance of mT5 on orm & pcm and gain improvements over baseline Rouge scores for the other languages we consider, with yor benefiting the most.

#### 4.1.4 Text Classification

Our results for the news classification task are presented in Table 1. We finetune AfriTeVa V2 on MasakhaNews for each language, framing it as a text–to–text task by predicting the class of each article in the decoding sequence and report results of 3 random seeds. On average, AfriTeVa V2 yields better F1 scores across all languages and has the best F1 score on 10 out of 16 languages.

### 4.2 Discussion

#### 4.2.1 Results for Nigerian Pidgin

AfriTeVa V2 does not outperform baselines for text classification, machine translation and summarization on Nigerian Pidgin (pcm). We note that AfriTeVa V2 was not pretrained on Nigerian Pidgin. As Nigerian Pidgin is an English-based creole, models pretrained on large amounts of English text are expected to be performant for the language. However, AfriTeVa V2 was pretrained on far less English text than the baselines we compare to, save for AfriTeVa. Still, we obtains results for Nigerian Pidgin that are competitive with the best baselines across the evaluation tasks.

#### 4.2.2 Impact of Data Quality on LMs

Previous works have shown the correlation between the quality of the data used in pretraining a model and the performance of the trained model (Rae et al., 2021; Kreutzer et al., 2022; Hernandez et al., 2022). AfriTeVa V2's improvement over baselines in downstream tasks suggests that this is true. We note that AfriTeVa V2 outperforms the

| Model | hau | ibo | orm | pcm | som | swa | yor | AVG |
|---|---|---|---|---|---|---|---|---|
| mT5 | 39.4/17.7/31.7 | 31.6/10.2/24.5 | **18.7/6.2/16.2** | 38.0/15.1/29.9 | 31.6/11.6/24.2 | 37.7/**17.9/30.9** | 31.7/11.7/25.1 | 32.7/12.9/26.1 |
| AfriTeVa V2 | **41.0/18.8/32.8** | **33.4/12.7/25.6** | 18.5/6.1/16.0 | 37.7/14.6/29.1 | **33.3/12.8/26.1** | **38.1**/17.8/**30.9** | **38.9/16.7/29.9** | **34.4/14.2/27.2** |

Table 3: XL-SUM results: Performance based on Rouge-1, Rouge-2 and Rouge-L. AfriTeVa V2 is generally more effective than mT5.

| Metric | Model | bem | hau | ibo | kin | twi | yor | zul | AVG |
|---|---|---|---|---|---|---|---|---|---|
| F1 | mT5 | 2.9 | 25.8 | 41.7 | 25.5 | **5.3** | 11.9 | 24.7 | 17.6 |
| | AfriTeVa-Base | 3.5 | 4.6 | 5.5 | 4.8 | 5.4 | 6.1 | 4.4 | 4.9 |
| | AfriMT5-Base | **6.4** | 39.7 | 40.7 | 30.3 | 5.3 | 21.8 | 31.9 | 25.2 |
| | AfriTeVa V2 | 5.7 | **45.4** | **57.1** | **45.4** | 2.1 | **37.6** | **45.9** | **34.2** |
| EM | mT5 | 1.1 | 22.3 | 34.7 | 20.2 | **3.5** | 7.8 | 20.9 | 13.9 |
| | AfriTeVa-Base | 2.0 | 2.7 | 4.2 | 3.2 | 3.1 | 3.9 | 3.1 | 3.2 |
| | AfriMT5-Base | 4.2 | 33.0 | 33.0 | 23.1 | 2.9 | 15.7 | 25.5 | 19.6 |
| | AfriTeVa V2 | **5.2** | **36.7** | **47.7** | **33.7** | 1.4 | **29.5** | **37.8** | **27.4** |

Table 4: Cross-lingual Question Answering Results: F1 and Exact Match (EM) Accuracy scores on the test set of AfriQA (Ogundepo et al., 2023). For both metrics, AfriTeVa V2 outperforms mT5 except for twi.

larger AfriMT5 & AfriByT5 (Alabi et al., 2022) which were trained on unfiltered mC4 corpus.

However, our pretraining dataset, WURA, contains ∼1.5× more data than mC4 contains across 16 African languages. Thus, more experiments are needed to separate the effects of scale from that of data quality.

## 5 AfriTeVa V2 Large Model

We also pre-train a large variant of AfriTeVa V2 using the same configuration of the T5-large model except for the vocabulary size which we set to be 150, 000, similar to the configuration of AfriTeVa V2 (base) as detailed in subsection 3.1. We present the effectiveness of scaling to a large model size on summarization and news topic classification tasks in Appendix C. [4]

## 6 Related Work

Absence of a large monolingual corpus has always been the major challenge of leveraging the benefits of self-supervised pretraining for building representation and language models for African languages. The most available corpus are mostly from religious corpus like Bible (Resnik et al., 1999) or JW300 (Agić and Vulić, 2019), Wikipedia and Common Crawl archive. The latter often has significant quality issues (Kreutzer et al., 2022).

Earlier works on building word representation models for African languages showed the importance of developing FastText embeddings with small high-quality data (Alabi et al., 2020) over pretrained FastText embeddings developed from noisier common crawl data. Obtaining such high-quality data is tedious since it involved curating several verified sources manually. Thus, previous works have prioritized filtering of the common crawl data to produce better quality dataset for pretraining (Conneau et al., 2020; Ortiz Suárez et al., 2019; Xue et al., 2021; Bapna et al., 2022). However, quality issues still persist in those filtered corpora. An alternative to this is basically aggregating high quality data for African languages mostly from verified sources (Ogueji et al., 2021; Leong et al., 2022; Palen-Michel et al., 2022). However, this often results in smaller sized corpus.

The current models with impressive performance on African languages simply aggregate both low-quality data and high-quality data for pretraining (Alabi et al., 2022; Adebara et al., 2022). The quality of these models implies that there must be significant portions of the data that are of good quality. To this end, we systematically and rigorously filtered these low-quality data from mC4 corpus for African languages, similar to the OSCAR dataset approach. [5] To the best of our knowledge, no previous work has done this. OSCAR dataset only has few documents for African languages e.g., 37.2MB for Afrikaans dataset while our filtered corpus has more than 4.5 GB.

## 7 Conclusion

In this work, we look to address the lack of large, quality pretraining dataset for African languages. While previous works have highlighted quality issues in existing pretraining dataset such as mC4, we demonstrate how these datasets can be enhanced by auditing their document sources and incorporating rigorous data filtering methods. To highlight the effectiveness of our approach and the relevance of this new dataset, we train a new T5 model, AfriTeVa V2, on our dataset. Our experiments show significant improvements across existing NLP benchmarks for African languages underscoring the impact of qualitative pretraining data in training language models.

---

[4]Due to space constraint, we include results in appendix.

[5]https://oscar-project.org/

## Limitations

The representativeness of our dataset poses a potential limitation. Despite our efforts to collect data from multiple African news websites, it is possible that our dataset does not fully capture the breadth and diversity of African news articles. The reliance on specific websites and the utilization of the mC4 dataset, along with existing corpora, may introduce inherent bias that our work does not address. Furthermore, our implementation of several-level filtering techniques, including the removal of non-linguistic content in the target language, does not guarantee the complete removal of all text in different languages or other toxic contents that may be present in the existing corpus.

Lastly, we acknowledge the need for future work to include more African languages. Our dataset only covers 16 languages, limiting the generalizability of our findings across the wide range of languages spoken in Africa.

## Acknowledgements

This research was supported in part by the Natural Sciences and Engineering Research Council (NSERC) of Canada and an AI for Social Good grant from the Waterloo AI Institute. Computational resources were provided by Compute Ontario and Compute Canada. We also thank the Google TRC program for providing us free cloud TPU access.

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

## A Data

### A.1 mC4 Audit

We aim to tease out heuristics that are guaranteed to help us quickly and reliably extract high-quality monolingual text across the African languages in mC4. First, we reduce the source URL of each document to its hostname[6] and keep a list of unique hostnames that exist for each language. For each language, we first sample a hostname then sample 20 documents sourced from the sampled hostname. This sampling strategy not only allows to audit more documents and sources faster, it allows us trace existing quality issues to the source URLs that produced the documents. We follow non-expert auditing strategies proposed by Kreutzer et al. (2022). Additionally, we also visit the hostname URL[7] to ascertain its purpose for speakers of the language and translate paragraphs in the document using Google Translate.

### A.2 Web Crawling

We open-source Otelemuye,[8] an extensible framework for large scale web-crawls. In our work, we crawl at a safe pace that does not degrade the website's performance and respect the rules websites publish in their robots.txt.[9] Where possible, we

---

[6]The hostname property of the URL interface is a string containing the domain name of the URL

[7]Some hostnames may have moved to new addresses or shut down permanently. In such cases, we check the Internet Archive.

[8]https://github.com/theyorubayesian/otelemuye

[9]https://developers.google.com/search/docs/crawling-indexing/robots/intro

| Sampling | Vocab Size | hau | ibo | kin | nya | sna | swa | xho | yor | zul |
|---|---|---|---|---|---|---|---|---|---|---|
| Config ① | 100,000 | 1.29 | 1.62 | 1.80 | 1.90 | 1.76 | 1.24 | 2.37 | 2.05 | 2.22 |
| | 150,000 | 1.25 | 1.53 | 1.67 | 1.74 | 1.64 | 1.21 | 2.20 | 1.97 | 2.06 |
| | 200,000 | 1.23 | 1.49 | 1.57 | 1.67 | 1.56 | 1.19 | 2.10 | 1.92 | 1.96 |
| | 250,000 | 1.22 | 1.47 | 1.54 | 1.63 | 1.53 | 1.19 | 2.03 | 1.90 | 1.91 |
| Config ② | 100,000 | 1.25 | 1.43 | 1.52 | 1.65 | 1.54 | 1.29 | 2.07 | 1.67 | 1.90 |
| | 150,000 | 1.21 | 1.39 | 1.43 | 1.51 | 1.45 | 1.25 | 1.94 | 1.59 | 1.77 |
| | 200,000 | 1.20 | 1.37 | 1.38 | 1.45 | 1.38 | 1.23 | 1.86 | 1.55 | 1.69 |

Table 5: **Tokenizer Fertilities**: We measure the fertilities of our tokenizers with varying vocabulary sizes using the MasakhanePOS dataset. The 150k tokenizer gives the best trade-off in size and fertility scores across all languages, especially in the second sampling configuration.

include the category under which each article was published. This information may be useful for identification of the domains in our dataset. We also release a list of the top document URLs for each language[10] and invite native speakers to audit these sources to help us improve the quality of WURA.

## B Tokenization

In multilingual settings, the design of tokenizers has great impact on the downstream utility and cost of inference of language models across languages (Petrov et al., 2023; Ahia et al., 2023). We characterize the performance of our tokenizers using *fertility* (Ács., 2019), defined as the number of subwords created per word (or per dataset) by the tokenizer. We compute fertility on the langauges covered by MasakhanePOS (Dione et al., 2023).

We train multiple unigram language models on our dataset using Sentencepiece (Kudo and Richardson, 2018) with vocabulary sizes ranging from $100,000$ to $250,000$. As shown in Table 6 above, our dataset sizes varies over orders of magnitude between languages. To alleviate unfair treatment of the lowest-resourced of the languages we consider, we follow Guillaume Lample and Alexis Conneau (2019) to learn the unigram language models on sentences sampled according to a multinomial distribution with probabilities $q_{i_{i=1...N}}$ calculated as follows:

$q_i = \frac{p_i^\alpha}{\sum_{j=1}^{N} p_j^\alpha}$ where $p_i = \frac{n_i}{\sum_{k=1}^{N} n_k}$ and $\alpha = 0.3$. $N$ denotes the number of languages and $n_i$, the number of sentences in language $i$. We denote this as sampling configuration ①. We also investigate a sampling configuration ② in which we further up-sample languages which still do not have adequate

---

[10]https://github.com/castorini/AfriTeVa-keji#dataset

| | African Languages in mC4 | | | | | |
|---|---|---|---|---|---|---|
| **Language** | **# Crawled Articles** | **# Wikipedia Articles** | **# mC4 Articles** | **# Combined Articles** | **# De-duped Articles** | **Size (GB) Articles** |
| Afrikaans (afr) | 139,977 | 107,860 | 978,740 | 1,226,577 | 1,158,680 | 4.8 |
| Amharic (amh) | 22,831 | 15,713 | 112,843 | 151,387 | 150,958 | 1.2 |
| Chichewa (nya) | — | 1,135 | 42,917 | 44,052 | 44,052 | 0.4 |
| Hausa (hau) | 247,507 | 25,957 | 147,028 | 420,492 | 399,866 | 0.9 |
| Igbo (ibo) | 6,196 | 16,158 | 34,802 | 57,156 | 57,095 | 0.2 |
| Malagasy (mlg) | 35,839 | 95,612 | 110,841 | 242,292 | 240,233 | 0.5 |
| Sesotho (sot) | — | 1,076 | 41,547 | 42,623 | 42,623 | 0.2 |
| Shona (sna) | 10,637 | 10,847 | 48,337 | 69,821 | 67,762 | 0.5 |
| Somali (som) | 585,928 | 11,241 | 513,028 | 1,110,197 | 1,084,982 | 2.3 |
| Swahili (swa) | 265,733 | 77,017 | 831,162 | 1,173,912 | 1,151,393 | 3.5 |
| Xhosa (xho) | — | 1,554 | 24,992 | 26,546 | 26,546 | 0.1 |
| Yoruba (yor) | 28,463 | 32,915 | 20,463 | 81,841 | 81,632 | 0.1 |
| Zulu (zul) | — | 11,331 | 61,387 | 72,718 | 72,718 | 0.7 |
| | African Languages not in mC4 | | | | | |
| Afaan Oromoo (orm) | 18,675 | 1,535 | — | 22,410 | 22,410 | 0.06 |
| Kinyarwanda (kin) | 17,218 | 7,423 | — | 32,437 | 32,437 | 0.10 |
| Tigrinya (tir) | 8,728 | 427 | — | 9,155 | 9,155 | 0.03 |
| Total | 1,393,097 | 422,536 | 2,968,087 | 4,793,623 | 4,652,549 | 18.9 |
| | Other Languages | | | | | |
| Arabic (arz) | — | 1,617,402 | — | 1,617,402 | 1,617,402 | 0.72 |
| English (eng) | 31,727 | 1,500,000 | — | 1,531,727 | 1,531,727 | 4.0 |
| French (fra) | 103,529 | 1,500,000 | — | 1,603,529 | 1,603,529 | 3.6 |
| Portuguese (por) | 107,670 | 1,102,551 | — | 1,210,221 | 1,210,221 | 2.3 |
| Total | 1,636,023 | 6,142,489 | 2,968,087 | 10,756,502 | 10,615,428 | 29.5 |

Table 6: **WURA Dataset Statistics**: We provide the count of crawled articles, Wikipedia articles, original mC4 articles, and final size before passage-level filtering for each language. In total, we have ∼4.7M articles, more than 1.5 times what mC4 contains across 16 African languages.

| Model | hau | ibo | orm | pcm | som | swa | yor | AVG |
|---|---|---|---|---|---|---|---|---|
| AfriTeVa V2 (Base) | 37.3/**16.3/29.6** | 22.6/8.1/17.7 | 16.1/**5.7**/14.1 | 37.0/14.5/29.1 | 29.3/**10.1/23.2** | 34.2/15.5/27.9 | 36.2/15.1/26.9 | 30.9/12.6/24.6 |
| AfriTeVa V2 (Large) | **38.1**/16.2/29.5 | **34.9/12.8/25.9** | **16.8**/5.2/**14.4** | **38.8/14.9/30.0** | **29.8**/10.0/23.1 | **38.5/18.1/31.4** | **38.2/16.0/27.6** | **34.2/13.9/26.7** |

Table 7: XL-SUM results: Performance based on Rouge-1, Rouge-2 and Rouge-L. AfriTeVa V2 Large outperforms AfriTeVa V2 Base across all languages considered.

| Model | amh | eng | fra | hau | ibo | lin | lug | orm | pcm | run | sna | som | swa | tir | xho | yor | AVG | AVG$^{SL}$ |
|---|---|---|---|---|---|---|---|---|---|---|---|---|---|---|---|---|---|---|
| AfriTeVa V2 (Base) | **92.8** | 90.6 | 88.0 | 89.4 | 86.1 | 86.0 | 91.1 | **90.8** | 96.8 | 92.3 | **93.3** | 75.7 | **87.0** | **86.4** | 93.6 | **92.3** | 89.5 | 88.9 |
| AfriTeVa V2 (Large) | 92.4 | **91.1** | **88.2** | **89.8** | **88.4** | **90.2** | **92.1** | 88.2 | **96.9** | **92.6** | 93.2 | **77.9** | 86.0 | 86.0 | **94.6** | 91.8 | **90.0** | **89.3** |

Table 8: MasakhaNews Classification Results: Evaluation is done using the weighted F1 score and the scores presented are averaged across 3 seeds. AfriTeVa V2 Large marginally improves overs Base results.

representation after sampling sentences with the calculated probabilities. Simply, after calculating probabilities using ①, we upsample by a factor of 10 for ibo, kin, nya, sna, sot, tir, xho, and a factor of 5 for amh, arz, mlg, som. We make this choice of upsampling factor taking into consideration the maximum amount of data we can train with given our CPU resources. The fertility of tokenizers trained on the sentences obtained by both

sampling configurations are presented in Table 5.

Across both configurations ① & ②, we obtain the best tradeoff between fertility distributions across the languages and vocabulary size at 150,000. Tokenizers obtained from ② perform better across board, improving fertility markedly for ibo, kin, nya, sna, xho, yor and zul without affecting fertility for hau and swa negatively.

## C AfriTeVa V2 Large

We also pretrain a large variant of AfriTeVa V2 and present its effectiveness on summarization (Table 7) and classification (Table 8). For summarization, we finetune both models for 10 epochs and make inference using beam search with width of 4. We gain improvements over the base model across both tasks, particularly for summarization where `ibo` benefits the most.