# OpenReview forum: "Better Quality Pre-training Data and T5 Models for African Languages"
_EMNLP/2023/Conference — EMNLP 2023 Main_

### Official Review · Reviewer_ckvB · 2023-07-25

**Soundness:** 3

**Excitement:**

3: Ambivalent: It has merits (e.g., it reports state-of-the-art results, the idea is nice), but there are key weaknesses (e.g., it describes incremental work), and it can significantly benefit from another round of revision. However, I won't object to accepting it if my co-reviewers champion it.

**Paper Topic And Main Contributions:**

In this paper, the authors introduce a new multilingual dataset comprising 16 African languages and high-resourced languages popularly spoken in Africa - Egyptian Arabic, English, French, and Portuguese. The dataset was created by: 1) extensively auditing and cleaning mC4, a very popular and commonly used multilingual dataset for pretrained models; 2) crawling websites across multiple domains such as news and Wikipedia. Further, the authors use their newly created dataset to pretrained a multilingual T5 model, and they evaluate its performance on multiple downstream tasks. The newly introduced model improves over existing multilingual models across 17 languages and four downstream tasks.

**Questions For The Authors:**

1. Are you planning on making the model and the dataset publicly available?

**Reasons To Accept:**

1. The paper introduces a new pretraining corpus for 16 low-resourced African languages. This is beneficial to the wider NLP community to help accelerate the development of newer models. The data creation section was thorough and detailed, highlighting the careful thinking that went into the data.
2. The authors demonstrate the effectiveness and the high quality of their newly created dataset by using it to pretrain a new T5 model that achieves better results than current existing mT5 models on a variety of tasks. The experiments are convincing and well-explained.

**Reasons To Reject:**

1. Some parts of the paper were confusing and hard to read:

   a) Section 1: In the intro, the authors mention that they introduce a dataset for 20 African languages. This is not precise as the dataset contains 16 African languages and four other languages that are spoken in Africa.

   b) Section 1 (L091): the authors mention that their model achieves better performance than mT5 models on 19 languages, but the abstract (L021) mentions 17 languages.

2. Reproducing the results when it comes to the downstream tasks is going to be hard, given that no discussion of hyperparameters is mentioned.

**Reproducibility:**

2: Would be hard pressed to reproduce the results. The contribution depends on data that are simply not available outside the author's institution or consortium; not enough details are provided.

**Reviewer Confidence:**

3: Pretty sure, but there's a chance I missed something. Although I have a good feel for this area in general, I did not carefully check the paper's details, e.g., the math, experimental design, or novelty.

**Typos Grammar Style And Presentation Improvements:**

Typos:
* L025: our new model more than doubles the performance --> Our new model achieves more than double the performance
* L054: news corpora
* L077: for the respective language --> for the respective languages
 * L094: our new model more than double the performance --> our new model achieves more than double the performance
* L112: We highlight three important findings of note of our own --> We highlight three important findings

Other comments:
* Table 4: F1 for hau should be bold.
* Why Table 5 calls the four non-African languages pivot languages? This is a specific language-transfer lingo, I would avoid it
* The use of pre-training vs pertaining is not consistent across the paper, I would stick to either one

---

> ### Author Rebuttal · Authors · 2023-08-28
>
> We thank reviewer ckvB for engaging with our work and providing thoughtful feedback. You found our curation techniques detailed, well-documented and thoughtful, and touted our work very relevant to the broader NLP community given our consistent improvement upon baselines across a diverse set of evaluation tasks.
>
> **Addressing your concerns**
>
> **Inconsistent claims and naming. Incorrect use of terms:** We thank you for pointing these out and take full responsibility. All the instances pointed out will be corrected and we will ensure consistency across the paper in our final submission.
>
> **Reproducibility questions:** We plan to release both the data and model artefacts to support further research. Our data cleaning pipeline as well as other code artifacts will be open-sourced. For downstream tasks, we evaluated our T5 model under the same conditions as the baselines to which we compared. Our camera-ready submission will also include an appropriate description of hyperparameters used in downstream tasks. All of this will ensure the reproducibility of our work.

---

### Official Review · Reviewer_A7Gh · 2023-08-04

**Soundness:** 2

**Excitement:**

3: Ambivalent: It has merits (e.g., it reports state-of-the-art results, the idea is nice), but there are key weaknesses (e.g., it describes incremental work), and it can significantly benefit from another round of revision. However, I won't object to accepting it if my co-reviewers champion it.

**Paper Topic And Main Contributions:**

This paper introduces a new multilingual corpus for 16 African languages and four pivot languages by combining (1) audited and cleaned mC4 (2) crawled indigenous websites and (3) Wikipedia. The empirical evaluation on four cross-lingual tasks suggests that the model pre-trained on the new corpus significantly outperforms mT5 (pre-trained on the original mC4) and AfriTeVa (pre-trained on small high-quality 1GB data).

**Reasons To Accept:**

1. a new corpus with a higher proportion of African data.
2. Superior performance of the new T5 model pre-trained on this corpus, compared to mT5 (pre-trained on the original mC4) and AfriTeVa (pre-trained on small high-quality 1GB data).

**Reasons To Reject:**

1. My main concern is: though the model achieves better performance, I cannot find sufficient evidence to support if the improvement truly comes from, well, "better quality".
2. I suspect that the improvement might be caused by the larger scale. For instance, because the mC4 takes the majority of the corpus according to Table 5, the authors find more available data than the original mC4 and they even update the crawls up to the current day (line 152~156).  However, I cannot find any ablation test to support that the filtering process actually works.
3. More comparisons of corpus statistics of African languages among the new corpus, the original m4c and 1GB data (AfriTeVa) are needed.
4.  The comparison is not fair. At least, the author should provide additional AVG of excluding those languages not in mC4.


**Reproducibility:**

2: Would be hard pressed to reproduce the results. The contribution depends on data that are simply not available outside the author's institution or consortium; not enough details are provided.

**Reviewer Confidence:**

3: Pretty sure, but there's a chance I missed something. Although I have a good feel for this area in general, I did not carefully check the paper's details, e.g., the math, experimental design, or novelty.

---

> ### Author Rebuttal · Authors · 2023-08-28
>
> We thank reviewer A7gh for engaging with our work and acknowledging that we introduced a useful dataset and improved upon strong baselines across a diverse set of evaluation tasks.
>
> **Addressing your concerns**
>
> **Sufficient evidence of improvement in performance:** We acknowledge your concern regarding evidence that our better quality data actually translates to improved model performance across the chosen evaluation tasks. While this concern may be laid to rest by ablation experiments in which we pretrain a language model on the unfiltered mC4 corpus, we were constrained by available resources at the time of submission. Fortunately, we have secured additional TPU Research Cloud Credits to support our work and will begin these ablation experiments. We also note that our model outperforms AfriMT5 which was trained on the unfiltered mC4 corpus by Adelani et al. Beyond this, as generative models take centre stage, the benefits of our cleaned corpora extend beyond model performance to solving the problem of model toxicity at the data level.
>
> **For fair comparisons, results should include mC4 averages without unseen languages** This is a fair criticism of our work. We agree that including averages that exclude language unseen by mT5 and other baseline models during pretraining is worthwhile. Note that since our corpus builds upon mC4 which mT5 & AfriMT5 were trained on, Afaan Oromoo, Kirundi and Tigrinya are the only languages present in our pretraining data but not theirs. Thus, averages without unseen languages do not change the conclusions of our work. For text classification on seen languages alone, ArawaT5 averages 88.9 F1 points compared to mT5’s  78.6. When translating from English into languages mT5 was pretrained on, ArawaT5 averages 18.0 BLEU compared to mT5’s 12.5, AfriMT5’s 14.0 and AfriByT5’s 15.9.  We observe a similar trend when translating into English, as well as for our summarization rouge scores.
>
> **Comparisons of corpus statistics (AfriTeVa vs. Wura):** A major aspect of work is our meticulous audit and extension of the mC4 corpora and this is why we compare the statistics of mC4 versus the dataset we introduce, Wura. AfriTeVa was pretrained on the 1 GB AfriBERTa dataset introduced by Ogueji et al. While we are willing to include an additional table that makes this comparison, we note that the AfriBERTa corpus is a paragraph-level dataset not a document-level dataset level like mC4 and Wura. As such, we can only compare on the basis of the disk size of data available per language.

---

### Official Review · Reviewer_Rqmg · 2023-08-04

**Soundness:** 4

**Excitement:**

5: Transformative: This paper is likely to change its subfield or computational linguistics broadly. It should be considered for a best paper award. This paper changes the current understanding of some phenomenon, shows a widely held practice to be erroneous in someway, enables a promising direction of research for a (broad or narrow) topic, or creates an exciting new technique.

**Paper Topic And Main Contributions:**

The work introduces a new unannotated corpus for 16 African languages called WURA. For languages in question, WURA is better curated and contains 1.5 times more articles than its predecessors (mC4).

The dataset is used for a pre-training language model based on T5 architecture called AwawaT5. The proposed model outperforms previous general and African-centric language models for most settings with four diverse downstream tasks.

I’m impressed by the richness of contributions of this short paper: both in data curation and language modeling.


**Questions For The Authors:**

A. In the abstract, you write that the improvement was achieved for 17 languages. Based on which task do you make this claim? Which language not included in the set of 16 African languages benefited?


B. Ad passage-level processing: how did you identify the potentially offensive chunks?


C. Why did you not compare to other baselines for QA and summarization tasks?

D. Suggestion: It would be interesting to compare the effectiveness of ArawaT5 in cross-lingual transfer.



**Reasons To Accept:**

Well-motivated study and highly needed in the community work on improving language resources for low and mid-resourced African languages.

The curation method is ingenious and includes multiple well-documented steps aimed at getting improved data representation for the languages in question.

The authors not only introduce a new dataset but also trains a model outperforming a set of strong baseline models comprising both general multilingual models and the models centered on African languages. The gains are manifested for diverse tasks (QA, text classification, translation, summarization) and are especially impressive on AfricaQA (+100% relative improvement in F1).


**Reasons To Reject:**

QA and summarization results are only compared against the mT5 model, ignoring the other baselines. The reason for that is not given in the paper.

**Reproducibility:**

4: Could mostly reproduce the results, but there may be some variation because of sample variance or minor variations in their interpretation of the protocol or method.

**Reviewer Confidence:**

4: Quite sure. I tried to check the important points carefully. It's unlikely, though conceivable, that I missed something that should affect my ratings.

**Typos Grammar Style And Presentation Improvements:**

Table 1: I suggest grouping the columns with results for African and other languages.

L179: Indicate that Table 5 is in the appendix. Also, here the style is inconsistent only the number is part of the link. I personally prefer this style of referring tables and figures to the one used in other places.

L281: ArawaT5 underperforms T5 also in Pidgin. The results for Swahili are on par.

---

> ### Author Rebuttal · Authors · 2023-08-28
>
> We thank reviewer Rqmg for engaging with our work and providing thoughtful feedback. You found our work well-motivated, and our curation detailed and well-documented. You also noted that the end result of our endeavour is a large, clean corpus beneficial for the wider NLP community and improved T5 models for African languages.
>
> **Addressing your concerns**
>
> **ArawaT5 underperforms T5 in Pidgin:** Indeed, ArawaT5 does not outperform baselines for text classification and summarization on Nigerian Pidgin but we remain competitive, especially on the summarization task! We note that ArawaT5 was not pretrained on Nigerian Pidgin. As Nigerian Pidgin is an English-based creole, models pretrained on large amounts of English text are expected to be performant for the language. However, ArawaT5 was pretrained on far less English text than the baselines we compare to, save for AfriTeVa. Still, our results for Nigerian Pidgin remain competitive with the best baselines across the evaluation tasks.
>
> **Comparison of QA and summarization results with other baselines:** We acknowledge that comparing our models with other models beyond mT5 is important. We have indeed run these experiments for QA and find that our model outperforms AfriTeVa and AfriMT5 by at least 9 F1 points on average.
> | Metric               | Model            | bem | hau  | bo  | kin  | twi | yor  | zul  | AVG           |
> |----------------------|------------------|--------------|---------------|---------------|---------------|--------------|---------------|---------------|---------------|
> | F1  | mT5              | 2.9          | 25.8          | 41.7          | 25.5          | 5.3 | 11.9          | 24.7          | 17.6          |
> |                      | AfriTeVa-Base    | 3.5          | 4.6           | 5.5           | 4.8           | **5.4**          | 6.1           | 4.4           | 4.9           |
> |                      | AfriMT5-Base     | **6.4** | 39.7          | 40.7          | 30.3          | 5.3          | 21.8          | 31.9          | 25.2          |
> |                      | ArawaT5 | 5.7          | **45.4**   | **57.1** | **45.4** | 2.1          | **37.6** |**45.9** | **34.2** |
> | EM  | mT5              | 1.1          | 22.3          | 34.7          | 20.2          | **3.5** | 7.8           | 20.9          | 13.9          |
> |                      | AfriTeVa-Base    | 2.0          | 2.7           | 4.2           | 3.2           | 3.1          | 3.9           | 3.1           | 3.2           |
> |                      | AfriMT5-Base     | 4.2          | 33.0          | 33.0          | 23.1          | 2.9          | 15.7          | 25.5          | 19.6          |
> |                      | ArawaT5 | **5.2** | **36.7** | **47.7** | **33.7** | 1.4          | **29.5** | **37.8** | **27.4** |
>
> Our summarization results using AfriMT5 and AfriTeVa will also be included in the final version to ensure that researchers can properly contextualize the results we obtain using ArawaT5.
>
> **Questions from reviewer Rqmg:**
> - **In the abstract, you write that the improvement was achieved for 17 languages. Based on which task do you make this claim? Which language not included in the set of 16 African languages benefited?:**
> As previously mentioned, we did not include Nigerian-Pidgin and this gives a total of 16. This mistake will be corrected in the camera-ready version of our paper.
> - **At passage-level processing: How did you identify the potentially offensive chunks?**
> For this work, our cleaning pipeline filters out any passages that contain at least one toxic word included in Facebook’s [Toxicity-200](https://github.com/facebookresearch/flores/tree/main/toxicity) dataset for the relevant language. While Kreutzer et al’s audit of mC4 found that mC4 did not contain a significant amount of offensive content (0.06% of sentences they audited) and our web crawls mainly focused on verified news publications, we feel strongly that this toxicity filtering is a necessary component of our data cleaning pipeline since we plan to expand this dataset over time.
>
> **Suggestion: It would be interesting to compare the effectiveness of ArawaT5 in cross-lingual transfer.**
>
> Thank you for the suggestion. Cross-lingual transfer is one of the reasons we included Arabic, English, French and Portuguese in our pretraining data. We believe this will allow zero-shot evaluation of ArawaT5 on tasks like information retrieval where African languages have little or no data. Also, there are a number of languages (such as Fon, Fufulde etc.) not included in our corpus because they do not have sufficiently large data. We plan to explore how well ArawaT5 generalizes to these languages and tease out the limits and tradeoffs of multilingualism as we increase the number of languages supported by our models.

---

### Meta-Review · Area_Chair_mzJV · 2023-09-19

**Recommendation:** 3

**Metareview:**

This paper creates a high-quality pretraining corpus in African languages from mC4, pretrains a Transformer-based language model, and further evaluated on multiple downstream tasks. The paper has a clear beneficial contribution to the community, not only in the trained model but also in cleaned corpus and preprocessing strategies. However, some straightforward improvement points of the current submission are identified by the reviewers such as fair comparison (e.g., "results should include mC4 averages without unseen languages"), and presentation clarity (e.g., the language code "pcm" should be explicitly mentioned as Nigerian Pidgin either in the main text or the appendix). During the rebuttal, further constructive comments on the analysis of the "quality" of the pretraining corpus have been brought up, especially since the paper is focused on auditing and cleaning mC4, but less ablation study is conducted. Nevertheless, the paper contributes to low-resource and underrepresented languages, especially on the creation of a new and better quality corpus along with empirical comparison against commonly-used multilingual models. We believe that this contribution is beneficial for the NLP community. Based on the reviews and rebuttal, the mmends accepting this paper, but the authors are highly encouraged to address the comments by Reviewer ckvB and Reviewer A7Gh.

---

### Decision · Program_Chairs · 2023-10-07

**Decision:**

Accept-Main

**Comment:**

This paper creates a high-quality pretraining corpus in African languages from mC4, pretrains a Transformer-based language model, and further evaluated on multiple downstream tasks. The paper has a clear beneficial contribution to the community, not only in the trained model but also in cleaned corpus and preprocessing strategies. However, some straightforward improvement points of the current submission are identified by the reviewers such as fair comparison (e.g., "results should include mC4 averages without unseen languages"), and presentation clarity (e.g., the language code "pcm" should be explicitly mentioned as Nigerian Pidgin either in the main text or the appendix). During the rebuttal, further constructive comments on the analysis of the "quality" of the pretraining corpus have been brought up, especially since the paper is focused on auditing and cleaning mC4, but less ablation study is conducted. Nevertheless, the paper contributes to low-resource and underrepresented languages, especially on the creation of a new and better quality corpus along with empirical comparison against commonly-used multilingual models. We believe that this contribution is beneficial for the NLP community. Based on the reviews and rebuttal, the mmends accepting this paper, but the authors are highly encouraged to address the comments by Reviewer ckvB and Reviewer A7Gh.